

# Effects of commercial beverages on the neurobehavioral motility of *Caenorhabditis elegans*

Wenjing Zhang[1], Nan Zhang[1], Shan Zheng[1], Wei Zhang[1], Jingjing Liu[1], Liwei He[1], Anastasia Ngozi Ezemaduka[2], Guojun Li[1,3], Junyu Ning[1,3], Bo Xian[4] and Shan Gao[1]

[1] Beijing Center for Disease Prevention and Control, Beijing Key Laboratory of Diagnostic and Traceability Technologies for Food Poisoning, Beijing, China
[2] Key Laboratory of Wetland Ecology and Environment, Northeast institute of Geography and Agroecology, Chinese Academy of Sciences, Changchun, China
[3] School of Public Health, Capital Medical University, Beijing, China
[4] School of Medicine, University of Electronic Science and Technology of China, Chengdu, China

Corresponding authors
Bo Xian, xianbo@uestc.edu.cn
Shan Gao, gaoshan1010@tom.com

## ABSTRACT

To study the effects of different types of commercially available drinks/beverages on neurobehavior using the model organism *C. elegans*, and critically review their potential health hazards. Eighteen kinds of beverages from the supermarket were randomly selected and grouped into seven categories namely functional beverage, tea beverage, plant protein beverage, fruit juice beverage, dairy beverage, carbonated beverage and coffee beverage. The pH value, specific gravity and osmotic pressure were also examined. The L4 stage N2 worms were exposed to different concentration of tested beverages (0, 62.5, 125, 250 and 500 $\mu$L/mL) for 24 h to measure the survival rate and locomotory behavior such as head thrashing, body bending as well as pharyngeal pumping. All the 18 beverages tested did not induce any visible lethal effects in the nematodes. However, exposure to different types of tested beverages exhibited different effects on the behavioral ability of *C. elegans*: (1) sports functional beverage and herbal tea drink accelerated the head thrashing and body bending of nematodes when compared to the control group ($P < 0.05$). (2) The vibration frequency of the pharyngeal pump of nematodes was significantly accelerated after treated with three plant protein beverages (almond milk, coconut milk and milk tea) and dairy products A and B ($P < 0.05$), and decelerated after treatment with other tested beverages. (3) Carbonated beverage significantly inhibits the head thrashing, body bending and pharyngeal pumping vibration ($P < 0.05$). Our results indicate that 18 kinds of popular beverages in the market have different influence on the neurobehavior in *C. elegans*, which may be related to their different components or properties. Further research would be required to conduct a systematic analysis of the effect of beverages by appropriate kinds, taking into consideration other endpoints such as reproduction, lifespan and molecular stress response, *etc.*, and to elucidate the mechanism for its potential health hazards.

## INTRODUCTION

Beverage refers to the products that are packaged in fixed quantity, directly or mixed with water, and the alcohol content does not exceed 0.5% by mass. The first commercially available beverage, "Moxie", was produced and found in a pharmacy in USA in 1884. In the past century, due to rapid development and innovation of high throughput technology and marketing system, beverage has developed from a local pharmaceutical product to a worldwide industry, and its consumption has increased rapidly all over the globe (*Tahmassebi & BaniHani, 2020*). As such, beverage industries have been shown to have positive impacts on tax revenue, driving industrial development and increasing employment.

According to the "general principles for beverages" (GBT 10789-2015) (*Standard China, 2015*), beverages are classified into eleven categories which includes packaged water, fruit and vegetable juices, protein drinks, carbonated drinks (soft drinks), special purpose drinks, flavor drinks, tea drinks, coffee drinks, plant drinks, solid drinks and other drinks. Beverages provide an important way for human to replenish water. They are also a source of intake of essential minerals and energy, with different types of beverages containing varied vitamin, mineral, protein, energy and so on. However, several studies have shown that there is a close relationship between the consumption of beverages and human health, especially in children and young people. Studies from many countries have shown that excessive consumption of beverages is associated with obesity, type 2 diabetes, osteoporosis, caries, asthma, depression, anxiety and other diseases (*Brown et al., 2005*; *Abid et al., 2009*; *Hu & Malik, 2010*). We proposed in this study to investigate the effect of beverages on the behavior of *C. elegans*.

*C. elegans*, a free-living coelomic organism, has emerged as excellent model in toxicological research, due to simplicity of maintenance, short life cycle, small body size and large brood size (*Harris et al., 2004*). It is the first multicellular organisms to have its genome completely sequenced. The genes and signaling pathways between *C. elegans* and human are highly conserved, thus provide animal data based on complete and metabolic activity, digestive, reproductive, endocrine, sensory and neuromuscular systems. It has great potential as a model for evaluating human related toxic effects (*Hunt, 2016*). In the late 1990s, nematodes were widely used in environmental toxicology research, and many sublethal endpoints have been developed, including the use of transgenic strains with specific biomarkers, growth and reproduction, pharyngeal pump feeding behavior and movement (*Mutwakil et al., 1997*; *Anderson, Boyd & Williams, 2010*; *Anderson, Cole & Williams, 2004*; *Höss & Weltje, 2007*; *Kurauchi, Morise & Eki, 2009*; *Boyd et al., 2010*). *Wang & Xing (2008a)* systematically studied the changes of head thrash, body bend and basic motion (forward and reverse motion in sinusoidal pattern) of nematodes after exposure to metals, and pointed out that a rapid and economic system can be established through consideration of endpoints such as head thrash, body bend and sinusoidal motion to evaluate the acute toxicity after exposure to heavy metals. In addition, nematode is the only organism that has a complete mapping of all neuronal species and interactions (*Bargmann, 1998*).

At present, laboratory research usually uses rodents to analyze intestinal toxicity, which costs a lot of time and money. Nematode is a coelenterate model animal, of which digestive system has many similarities with mammals, including acidified lumen, microvilli forming brush boundary, secretion of digestive enzymes, absorption and peristalsis of digestive components (*Hall & Altun, 2008*; *Chauhan et al., 2013*; *Katrin et al., 2015*). Mammals and *C. elegans* are comparable to the intestinal morphological abnormalities of cadmium and toxic lectins (*Hunt, Olejnik & Sprando, 2012*; *Katrin et al., 2015*). However, in contrast to mammals, intestinal toxin induced damage can also be observed in living nematodes by optical microscopy or COPAS (*Hunt, Olejnik & Sprando, 2012*). The nematodes are sucked into the pharyngeal to eat through pumping and peristalism. Therefore, it can be used as an oral toxicity model by adding a sample to the nutrient supply. In addition, in the presence of several known mammalian neurotoxins, the pharyngeal pumping rate is reduced, and can be observed in a single worm by optical microscope (*Boyd, McBride & Freedman, 2007*).

We chose 18 different types of commonly available beverages and examined their effects on head thrashing, body bending and pharyngeal pumping frequency in *C. elegans* to determine the potential effect of beverages on neurobehavior. We also determined their safe intake volume of consumption and provide insight into organisms' responses to the effects of consumable beverages and potential health hazards.

## MATERIALS & METHODS

### Chemical reagents and nematodes
NaCl, KCl, NaClO, NaOH, $CaCl_2$, $MgSO_4$, $K_2HPO_4$, $KH_2PO_4$, cholesterol, bacterial peptone, agar, tryptone, yeast extract, PI were all obtained from Sigma-Aldrich; the wild-type N2 *C. elegans* used in this study was originally obtained from the *Caenorhabditis* Genetic Center (Minneapolis, MN, USA). NGM medium, s-basal solution and k-medium solution were used in the experiment.

### Test substance
Eighteen beverages were purchased from the supermarket (Walmart), which were divided into seven categories, namely functional beverage, tea beverage, plant protein beverage, fruit juice beverage, dairy beverage, carbonated beverage and coffee beverage (Table 1). Samples were separated and stored at 4 °C.

### *C. elegans* maintenance
Nematodes were maintained on the nematode growth medium (NGM) plates seeded with *Escherichia coli* OP50 at 20 °C as described (*Brenner, 1974*). Gravid nematodes were washed off the plates into centrifuge tubes, and lysed with a bleaching mixture (0.45 mol/L NaOH, 2% HOCl). Age synchronous populations of L1- or L4-larvae were obtained as described (*Donkin & Dusenbery, 1993*).

### Grouping and processing
A liquid culture system was used in the experiment; four gradient doses were set for each beverage: ultrahigh dose group (500 µL/mL), high dose group (250 µL/mL), middle dose

**Table 1** Sample information.

| No. | Beverage category | Sample name | pH | Specific gravity | Osmotic pressure (Pa) |
|---|---|---|---|---|---|
| 1 | Fruit juice | mixed juice | 3.67 | 1.049 | 789 |
| 2 | Fruit juice | Single juice | 3.28 | 1.042 | 504 |
| 3 | Carbonated drink | Brown carbonated beverage | 2.37 | 1.042 | 740 |
| 4 | Carbonated drink | Colorless carbonated beverage | 3.11 | 1.046 | 709 |
| 5 | Carbonated drink | Orange carbonated beverage | 2.95 | 1.040 | 569 |
| 6 | Functional beverage | Sports functional drink | 2.90 | 1.019 | 244 |
| 7 | Functional beverage | Fatigue relieving functional drink | 3.42 | 1.043 | 496 |
| 8 | Tea beverage | Black tea beverage | 3.01 | 1.038 | 405 |
| 9 | Tea beverage | Green tea beverage | 6.05 | 1.015 | 145 |
| 10 | Tea beverage | Herbal tea drink | 5.48 | 1.030 | 281 |
| 11 | Coffee drink | Coffee drink | 6.67 | 1.041 | 404 |
| 12 | Plant protein beverage | Almond drink | 7.40 | 1.029 | 299 |
| 13 | Plant protein beverage | Coconut drink | 7.00 | 1.028 | 272 |
| 14 | Plant protein beverage | Milk tea beverage | 6.72 | 1.040 | 341 |
| 15 | Dairy beverage | Prepared milk beverage A | 4.10 | 1.026 | 302 |
| 16 | Dairy beverage | Prepared milk beverage B | 4.25 | 1.032 | 310 |
| 17 | Dairy beverage | Prepared milk beverage C | 4.15 | 1.030 | 341 |
| 18 | Dairy beverage | Prepared milk beverage D | 4.21 | 1.025 | 258 |

group (125 $\mu$L/mL), low dose group (62.5 $\mu$L/mL); k-medium group (0 $\mu$L/mL) was set as blank control. Four parallel wells were set as repeats.

## Beverage character detection

The pH value, specific gravity and osmotic pressure of 18 kinds of beverage were determined. The osmotic pressure of beverage samples was determined using an ice point osmotic pressure gauge (OM819.C, Löser Messtechnik, Berlin, Germany), the pH value was determined using a digital acidometer (Leici PHS-33; Leico, Shanghai, China), and the specific gravity was determined by taking one mL of the sample with a pipette and weighing it with a balance (Mettler Toledo, ML204). The instrument instruction manuals were consulted for specific operations, and the values of the measurement results were recorded. For more information, see Table 1.

## Acute toxicity test

A 200 $\mu$L aliquot of test beverages was added into each well of a 96-well plate, which was subsequently loaded with 30 nematodes for each treatment. After exposure to 0 $\mu$L/mL, 62.5 $\mu$L/mL, 125 $\mu$L/mL, 250 $\mu$L/mL, and 500 $\mu$L/mL concentrations of beverages for 24 h, nematodes were analyzed under the dissecting microscope (Zeiss, Oberkochen, Germany). Nematodes were judged to be dead if they did not respond to stimulation of platinum wire picker. The red fluorescence of nematodes after adding PI staining was used as an auxiliary judgment standard of nematode death.

### Body bending frequency

After the nematodes were cultured in 96-well plate for 24 h, nematodes were sucked out of the liquid well, and transferred to NGM medium plate without food (*Escherichia coli* OP50), the number of body bend in 20 s was recorded. One bend of body was defined as a change in the direction of the nematodes corresponding to the posterior bulb of the pharynx along the *y* axis, assuming that nematode was traveling along the *x* axis. Twenty nematodes were counted for each treatment, three replicates were performed.

### Head thrashing frequency

After the nematodes were cultured in 96-well plate for 24 h, the nematodes were sucked out of the liquid well, and transferred to NGM medium plate containing no food (*Escherichia coli* OP50), 60 μL M9 buffer solution was dripped on the NGM plate. After one minute recovery, the number of head thrash was recorded within 20 s. One head thrash was defined as a change in the direction of bending at the mid body. Twenty nematodes were observed for each treatment, three replicates were performed.

### Pharyngeal pumping rate

The pharyngeal pumping rate reflected the food intake ability of *C. elegans*. After the nematode was cultured in 96 well plate for 24 h, nematodes were sucked out of the liquid well and put onto a plate covered with OP50. The plate was placed under a fluorescent stereoscope and magnified enough to observe the movement of pharyngeal pump clearly. Each nematode was videotaped for 30 s. Twenty nematodes were recorded for each treatment, assays were performed in triplicate.

### Statistical analysis

All experimental data were expressed as mean $\pm$ SD. WPS was used to establish the database and SPSS 21.0 software was used for statistical analysis. Univariate analysis of variance was used for comparison between groups, and the significance level was set as $P < 0.05$. For standardized comparison, the movement frequency of the beverages-treated nematodes was divided by the frequency of those in control groups to obtain the corresponding fold.

## RESULTS

### Sample character information and lethal effect of the 18 kinds of drinks on nematodes

Eighteen kinds of beverages from the supermarket were randomly selected and grouped into seven categories namely functional beverage (sports functional drink and fatigue relieving functional drink), tea beverage (black tea beverage, green tea beverage and herbal tea drink), plant protein beverage (almond milk, coconut drink, milk tea beverage), fruit juice beverage (mixed and single juice), dairy beverage (prepared milk beverage A–D), carbonated drinks (brown, colorless and orange carbonated beverages), and coffee drink (coffee). The pH value, specific gravity and osmotic pressure of each tested drink were also examined (Table 1).

To determine the possible adverse effects of 18 beverages on nematodes, we first examined the acute toxicity of the tested beverages in *C. elegans*, through exposure of

synchronized L4 stage worm to five concentrations (0 µL/mL, 62.5 µL/mL, 125 µL/mL, 250 µL/mL, 500 µL/mL) of tested beverages. LC50 is the most important parameter to evaluate the acute toxicity of chemicals in toxicological studies. Our preliminary experiments demonstrated that all tested 18 kinds of beverages at concentrations of 0 µL/mL, 62.5 µL/mL, 125 µL/mL, 250 µL/mL, 500 µL/mL did not affect the survival of nematodes after 24-h exposure (Table S1). This indicates that LC50 of all the 18 kinds of tested beverage/drink were higher than 500 µL/mL.

## Effects of the eighteen tested beverages on neurobehavior in *C. elegans*

We further sought to examine the biological effect of the tested beverages on animal's locomotion ability at these five selected concentrations (0 µL/mL, 62.5 µL/mL, 125 µL/mL, 250 µL/mL, 500 µL/mL). Given that head thrashing and body bending of nematodes are considered neurobehaviors in *C. elegans*, we then asked whether exposure to the eighteen tested beverages has effect on the nematode head thrashing and body bending. Remarkably, the data presented in Figs. 1A and 1B demonstrate that the eighteen kinds of tested beverages have varied effects on the locomotion ability in *C. elegans*. Seven out of the 18 beverages (mixed juice, single juice, brown carbonated beverage, colorless carbonated beverage, fatigue relieving functional drink, prepared milk beverage C and D) reduced the frequency of head thrash, while four of them (sports functional drink, green tea, milk tea, prepared milk beverage A) showed the opposite effect of increased head thrash rate. Meanwhile, eight out of all the beverages (mixed juice, single juice, brown carbonated beverage, colorless carbonated beverage, fatigue relieving functional drink, green tea, coffee, prepared milk beverage C) inhibited the frequency of body bending, while nine of them (sports functional drink, black tea, herbal tea, almond milk, coconut drink, milk tea beverage, prepared milk beverage A, B and D) accelerated the body bending frequency (Figs. 1A and 1B; Tables S1, S3).

Furthermore, we examined the effects of the tested beverages on pharyngeal pumping, and our results showed that after exposure of the L4 nematodes larvae to tested beverages for 24 h, the frequency of nematodes pharyngeal pumping in most groups was affected. The eight beverages including mixed juice, single juice, brown carbonated beverage, colorless carbonated beverage, orange carbonated beverage, fatigue relieving functional drink, green tea, black tea, herbal tea, coffee, prepared milk beverage D reduced the pharyngeal pumping frequency, while the five beverages (almond milk, coconut drink, milk tea beverage, prepared milk beverage A and B) enhanced the rate of pharyngeal pumping in worms (Fig. 1C, Table S4).

## Locomotive behaviors are promoted by sports functional beverage and herbal tea drink

The changes in motor behavior of nematodes after beverage treatment can indirectly reflect the changes of nervous system. The evaluation indexes of nematode movement behavior mainly include average movement rate, head thrash, body bend, forward thrash, backward thrash and Omega/U thrash, among which head thrash and body bend frequencies are the most common indexes (*Williams & Dusenbery, 1990*; *Tsalik & Hobert, 2003*; *Wang &*

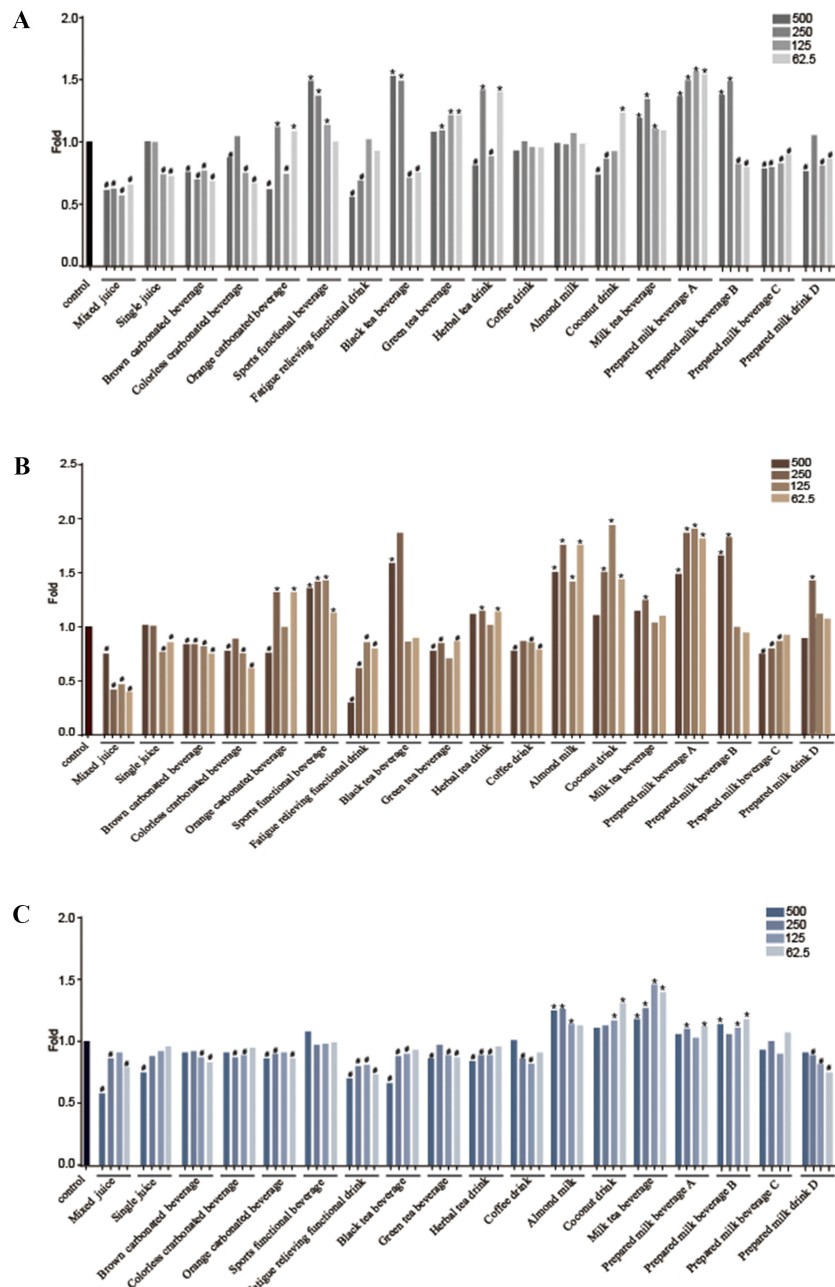

**Figure 1 Beverages exposure affect locomotion frequency in *C. elegans*.** The *x-axis* represents the different dose groups of 18 kinds of drinks, and the *y-axis* represents the ratio of the locomotive frequency of nematodes in each group to that in control group (fold = treated group/control group). (A) An asterisk (*) means head thrash frequency in treated group was higher than that in control group and $P < 0.05$, # means head thrash frequency in treated group was less than that in control group and $P < 0.05$. (B) An asterisk (*) means body bend frequency in treated group was higher than that in control group and $P < 0.05$, # means body bend frequency in treated group was less than that in control group and $P < 0.05$. (C) An asterisk (*) means pharyngeal pump frequency in treated group was higher than that in control group and $P < 0.05$, # means pharyngeal pump frequency in treated group was less than that in control group and $P < 0.05$.

*Wang, 2008b*). According to the experimental results, the sports functional beverage can accelerate nematode head thrashing from $60.07 \pm 7.27$ in the control group to $89.60 \pm 19.53$ (500 µL/mL), $82.00 \pm 17.16$ (250 µL/mL), $67.87 \pm 21.55$ (125 µL/mL), and body bending from $4.53 \pm 1.33$ in the control to $6.17 \pm 1.97$ (500 µL/mL), $6.43 \pm 2.05$ (250 µL/mL), $6.50 \pm 2.69$ (125 µL/mL) as well as $5.10 \pm 1.45$ (62.5 µL/mL), respectively ($P < 0.05$) (Figs. 2A, 2B). With reference to the nutritional composition table on the packaging bottle (Table S5), this kind of beverage takes the "vitamin group combination" as the market demand and adds four vitamins such as vitamin C, vitamin B3, vitamin B6 and vitamin B12, which can reduce the oxidative damage of the body, and participate in the body energy and protein metabolism, which in turn may enhance the movement ability of the body. The results obtained in herbal tea drink (HT) treated nematodes were similar to those in sports functional beverage which also promotes the body bend frequency from $6.53 \pm 1.36$ in the control group to $7.33 \pm 1.63$ (500 µL/mL), $7.53 \pm 1.25$ (250 µL/mL), $6.67 \pm 2.02$ (125 µL/mL) and $7.47 \pm 1.63$ (62.5 µL/mL), increased a degree by 14~15% ($P < 0.05$) (Fig. 2C), this is consistent with the results of *Lin et al. (2020)*. In their study, when day 17 adult nematodes were treated with HT at concentrations of 10 mg/mL, 30 mg/mL and 50 mg/mL, the number of free movement nematodes increased significantly, and the number of individuals who could not complete body movement after stimulation decreased, indicating that HT can enhance the motility of nematodes.

## Three plant beverages as well as dairy products A and B accelerate pharyngeal pumping rate in *C. elegans*

There is correlation and interaction between the movement and feeding behaviors of nematodes. The pharynx is the main feeding organ of worm. Presentation of food triggers sensory inputs that drive an increased pumping that draws bacteria into the gut (*Dillon et al., 2016*). The feeding behavior of nematodes requires rapid head muscle swing to complete, and it also needs body movement to shorten the distance from food. *McKay et al. (2004)* measured nematode feeding after incubating wild-type N2 and a variety of Eat defects mutants in pumping stimulant (5-HT) and pumping inhibitor (arecoline) as well as exposure to environmental toxicants (cadmium, chlorpyrifos), thereby confirming that the pharynx of *C. elegans* is a favorable model to study neuromuscular function and effects of chemicals on neuromuscular activity (*i.e.,* eating), clarifying the authoritative position of pharyngeal pump in neurotoxicity evaluation. In this study, the effect of 18 kinds of beverages on the feeding behavior of nematodes was measured by the rate of pharyngeal pump movement.

According to the experimental results, the vibration frequency of the pharyngeal pump of nematodes was significantly accelerated after the worms were treated with three plant protein beverages (almond milk, coconut milk and milk tea) and dairy products A and B, while a decrease in the rate of pharyngeal pump was seen after the worms were treated with other tested beverages. As shown in Fig. 3A, almond milk increased the pharyngeal pump vibration frequency of nematodes by 14%~26%, from $41.40 \pm 7.18$ in the control group to $51.80 \pm 14.29$ (500 µL/mL), $52.00 \pm 7.24$ (250 µL/mL) and $47.18 \pm 9.59$ (125 µL/mL), respectively ($P < 0.05$); coconut milk accelerated pharyngeal pumping by 17%~31%, from
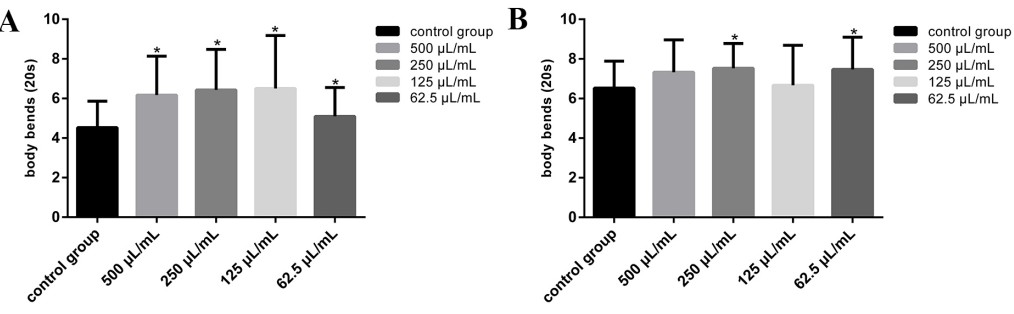

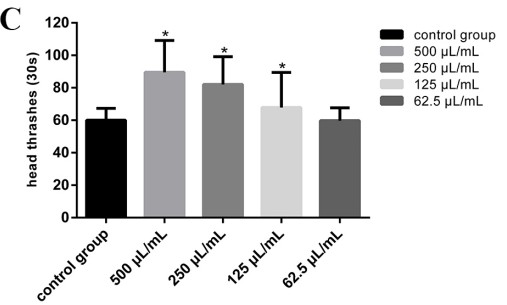

**Figure 2** **Sports functional beverage and herbal tea drink exposure affect head thrashing and body bending in *C. elegans*.** (A, B) The frequencies of head thrashes and body bends in 20 s of nematodes treated by sports functional beverage. (C) The frequencies of body bends in 20 s of nematodes treated by herbal tea drink. Data were presented as mean ± SD. * $P < 0.05$.

$41.40 \pm 7.18$ in the control group to $48.26 \pm 7.92$ (125 µL/mL) and $54.05 \pm 7.31$ (62.5 µL/mL), respectively ($P < 0.05$, Fig. 3B); the same change to milk tea which increased the frequency of pharyngeal pump by 18%~46%, from $41.40 \pm 7.18$ in the control group to $48.70 \pm 7.8$ (500 µL/mL), $52.50 \pm 7.29$ (250 µL/mL), $60.30 \pm 7.89$(125 µL/mL) and $57.85 \pm 9.18$ (62.5 µL/mL), respectively (Fig. 3C, $P < 0.05$). At the same time, prepared milk drinks A and B showed positive effect on the nematodes' pharyngeal pump by increasing the vibration frequency by 10%~12% and 11%~18% respectively ($P < 0.05$). In particular, prepared milk beverage A increased the frequency of pharyngeal pumping from $50.70 \pm 9.81$ in the control group to $55.75 \pm 6.45$ (250 µL/mL) and $56.70 \pm 7.78$ (62.5 µL/mL), whereas prepared milk beverage B increased the frequency of pharyngeal pumping from $50.70 \pm 9.81$ in the control group to $57.90 \pm 8.83$ (500 µL/mL), $56.40 \pm 8.02$ (125 µL/mL) and $59.75 \pm 6.34$ (62.5 µL/mL) ($P < 0.05$, Figs. 3D, 3E). It has been reported that nematodes respond to changes in metabolic and nutritional status through a variety of behavioral adaptations, including changes in food intake and food hunting, and the pharyngeal pump feeding behavior of nematodes involves both endogenous and exogenous neuromodulation (*Dillon et al., 2016*). Nematodes exhibit slow forward and backward movement in the presence of food, and generate a search state after being removed from food (*Wakabayashi, Kitagawa & Shingai, 2004*; *Cermak et al., 2020*). According to the data obtained in the current work, the plant protein beverage and dairy products A and B

induced the acceleration of pharyngeal pumping and body bending frequency at the same time in nematodes, while fruit juice, carbonated drink, fatigue relieving functional drink, tea and coffee all showed the characteristics of weakened feeding and body movement, which may be related to the food search process of nematodes, in other words, plant protein beverages as well as dairy products A and B treated nematodes were more likely to exercise actively to find and eat food. Furthermore, according to previous research, carbonated drinks can cause nerve and muscle function damage by disrupting the body's redox balance, and the motor procedures of nematodes, such as locomotion and feeding, are coupled by neuromodulators, which may be the cause of nematode feeding and movement weakness after exposure treatment to certain beverages (*El-Terras et al., 2016*; *Cermak et al., 2020*; *Zhu et al., 2022*). In the future, researchers may explore the possible causes through molecular mechanism research .

## Carbonated beverage inhibit motility and feeding in nematodes

It had been reported that drinking carbonated beverages for long periods of time can induce oxidative stress in brain tissue of Wistar rats, reduce the expression of antioxidant markers, and reduce the expression of acetylcholinesterase (AChE) in serum and brain tissue (*El-Terras et al., 2016*). According to Figs. 4A, 4B, 4C, 4D, the brown carbonated drink and colorless carbonated drink reduce the head thrash frequency and body bend frequency of nematodes by 23%~30% and 12%~32%, 15%~25% and 21%~38%, respectively. The brown carbonated drink reduced head thrashing from $79.60 \pm 15.15$ in the control group to $60.80 \pm 10.50$ (500 µL/mL), $55.33 \pm 6.44$ (250 µL/mL), $60.93 \pm 9.06$ (125 µL/mL), $54.97 \pm 10.71$ (62.5 µL/mL), and colorless carbonated drink reduced frequency from $79.60 \pm 15.15$ in the control group to $69.93 \pm 19.65$ (500 µL/mL), $59.33 \pm 4.43$ (125 µL/mL), $53.60 \pm 6.49$ (62.5 µL/mL). Body bend frequency was inhibited from $6.17 \pm 1.90$ in the control to $5.17 \pm 1.18$ (500 µL/mL), $5.20 \pm 1.21$ (250 µL/mL), $5.07 \pm 1.55$ (125 µL/mL), $4.60 \pm 1.38$ (62.5 µL/mL) by brown carbonated drink , and from $6.17 \pm 1.90$ in control to $4.83 \pm 1.97$ (500 µL/mL), $4.70 \pm 1.66$ (125 µL/mL), $3.80 \pm 1.40$ (62.5 µL/mL) by colorless carbonated drink. In addition, all three types of carbonated beverages can reduce pharyngeal pump vibration. The brown carbonated beverage reduced the vibration frequency of the pharyngeal pump by 13% from $65.50 \pm 6.92$ in the control group to $56.74 \pm 12.06$ (125 µL/mL) and $54.65 \pm 15.64$ (62.5 µL/mL) ($P < 0.05$, Fig. 4E); colorless carbonated beverages reduced the frequency by 11%~13% from $65.50 \pm 6.92$ in the control group to $56.75 \pm 5.67$ (250 µL/mL) and $58.05 \pm 10.99$ (125 µL/mL) ($P < 0.05$, Fig. 4F), and orange carbonated beverage reduced the vibration frequency by 10%~14% from $65.50 \pm 6.92$ in the control group to $56.10 \pm 8.63$ (500 µL/mL), $58.95 \pm 8.43$ (250 µL/mL) and $56.20 \pm 12.86$ (62.5 µL/mL) ($P < 0.05$, Fig. 4G). Previous research has linked REDOX imbalance in nematodes to muscle mass and function loss. Nematode muscle function is primarily manifested as pharyngeal pump (*Zhu et al., 2022*). Carbonated drinks are known to cause oxidative stress and after the expression of certain genes related to brain activity (*El-Terras et al., 2016*), which could explain why nematode's pharyngeal pump vibration frequency was inhibited in this study.

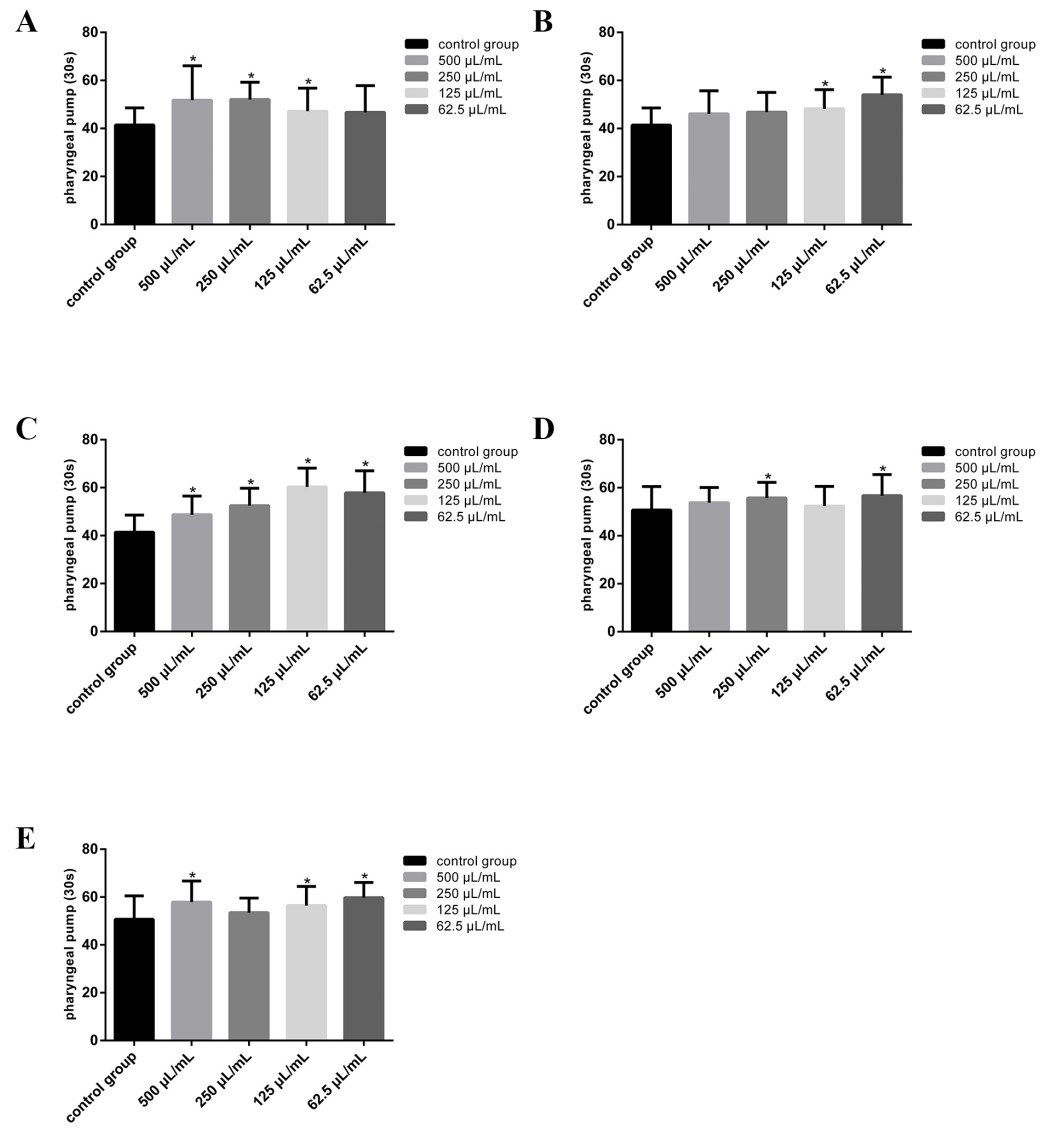

**Figure 3** **Beverages exposure affect pharyngeal pumping in *C. elegans*.** (A, B, C) The effect on pharyngeal pump frequency of nematodes treated by almond milk, coconut milk and milk tea beverage, respectively. (D & E) The effect on pharyngeal pump frequency of nematodes treated by Prepared milk beverage A and B. Data were presented as mean ± SD. * $P < 0.05$.

## DISCUSSION

In this study we report our findings on the effects of commonly available and consumed beverages/drinks on neurobehavior such as head thrashing, body bending and pharyngeal pumping in *C. elegans*. It is worth to note that many biological processes and signaling pathways of nematodes, such as survival, redox reaction, aging and neurodegenerative diseases, are conserved to those of human beings (*Hulme & Whitesides, 2011*), thus proving the nematode *C. elegans* an ideal model organism for biological experiments.

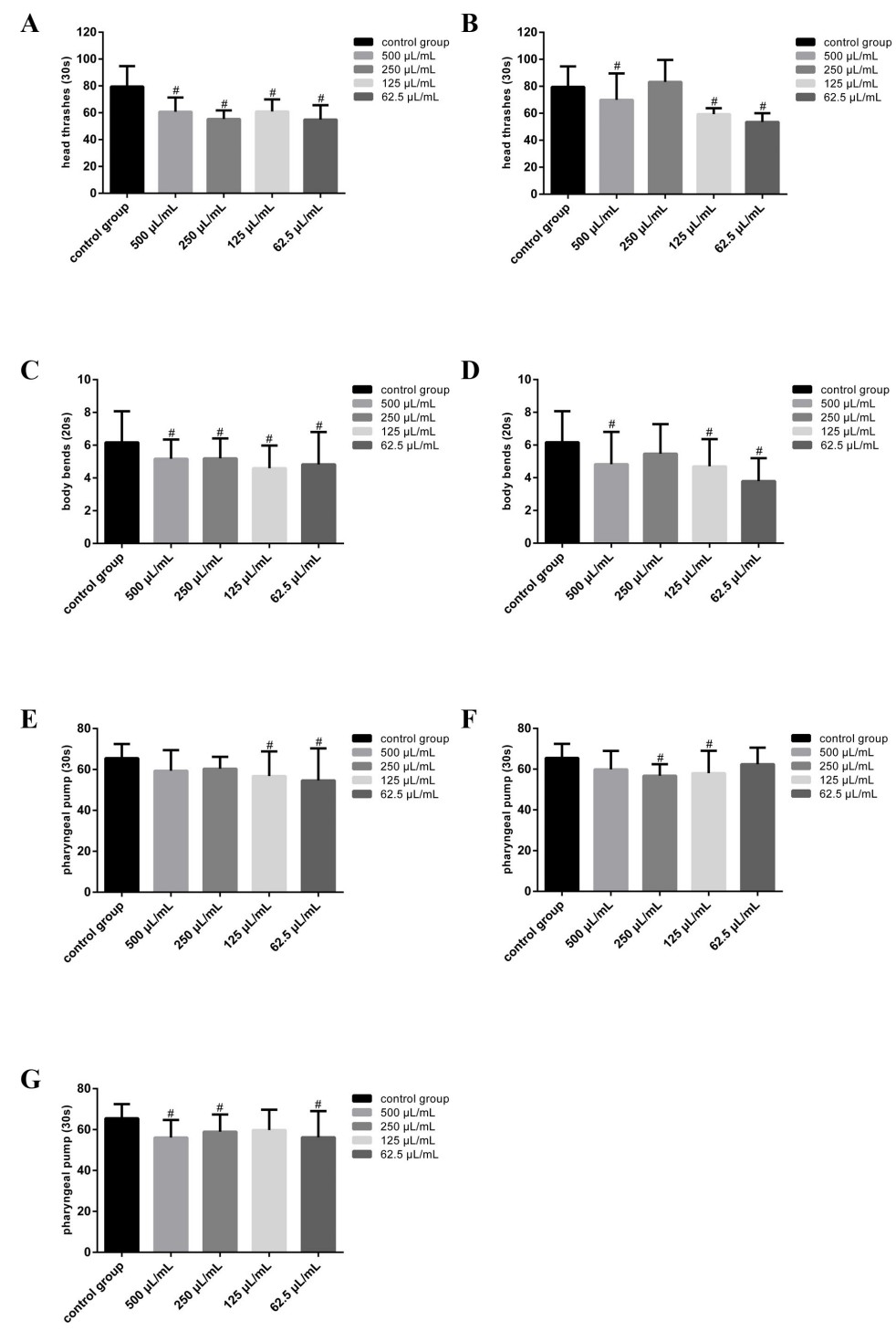

**Figure 4  Carbonated drinks exposure affect locomotion in *C. elegans*.** (A & B) The effect on head thrashes of nematodes treated by brown and colorless carbonated beverages. (C & D) The effect on body bends of nematodes treated by brown and colorless carbonated beverages. (E), (F & G) The effect on pharyngeal pump frequency of nematodes treated by brown, colorless and orange carbonated beverages. Data were presented as mean ± SD. # $P < 0.05$.

It has been reported that LC50 of nematode is conservative when compared with rats and mice, and the LC50 of nematode can be used to pre evaluate the acute toxicity of chemicals; and to an extent has certain accuracy to predict the acute toxicity of chemicals to rodents and even to human health risks (*Li et al., 2013*). In this study, we found that there was no acute lethal effect of the 18 kinds of beverages sold in the market on nematodes, but they had effects on indicators of motor behavior and feeding behavior such as head thrash, body bend and pharyngeal pump frequencies. Different types of drinks have different effects on the behavior of nematodes, which may be related to their different components or properties.

The nematode nervous system is made up of 302 neurons and control 143 muscle cells. To maintain board coordination, nematodes generate a series of well-defined motor programs (*e.g.,* locomotion, feeding, and postural changes of the head and body) that are controlled by specialized brain circuits acting on specific muscle groups. These motor circuits are linked together by a variety of neuromodulators, including dopamine, serotonin (5-HT), neuropeptide PDF, and others (*Cermak et al., 2020*). The movement behavior of nematodes is a common endpoint of sublethal toxicity (*Leung et al., 2008*). Nematodes moves in a sinusoidal style *via* alternative ventral and dorsal muscle contraction, of which the process is controlled by many kinds of neurons, nerve impulse conduction between neurons and neuromuscular interaction (*Omura et al., 2012*). Therefore, improper functioning of these processes *via* defect of the necessary organ(s) would lead to dyskinesia (*Loria, Hodgkin & Hobert, 2004*). According to our results, the sports functional beverage accelerated the head thrashing and body bending of nematodes. The 'vitamin group combination' in the sports functional beverage may reduce the oxidative damage of the body and participate in the energy metabolism and protein metabolism of the body, so as to enhance the movement ability. On the contrary, fruit juice, carbonated drinks, prepared milk C and fatigue relief functional drinks all reduced the frequency of head thrash and body bend, which may be related to the respective components of these drinks.

*C. elegans* eat bacteria *via* the pharynx, and availability of food triggers sensory inputs that drive an increased pumping that got bacteria into the gut (*Dillon et al., 2016*). Pharyngeal pump of nematode is a neuromuscular pump, which belongs to the feeding organ. According to the experimental results, the vibration frequency of the pharyngeal pump of nematodes was significantly accelerated after the worms were treated with three plant protein beverages (almond milk, coconut milk and milk tea) and dairy products A and B, and decelerated after treated with other tested beverages. It has been reported that nematodes respond to changes in metabolic and nutritional status through a variety of behavioral adaptations, including changes in food intake and food hunting, and the pharyngeal pump feeding behavior of nematodes involves both endogenous and exogenous neuromodulation (*Dillon et al., 2016*). Nematodes exhibit slow forward and backward movement in the presence of food, and native dopamine signaling in *C. elegans* is involved in driving the response (*Wakabayashi, Kitagawa & Shingai, 2004*; *Cermak et al., 2020*). According to the data, the plant protein beverage and dairy products A and B induced the acceleration of pharyngeal pump and bending frequency at the same time, while fruit juice, carbonated drinks, fatigue relieving functional beverages, tea and coffee all showed

the characteristics of weakened feeding and mobility, which may be related to the food search process of nematodes, that is, plant protein beverages as well as dairy products A and B treated nematodes were more likely to exercise actively to find and eat food.

Studies have shown that there is a link between consumption of soft drinks and depression, consuming a lot of drinks can increase the risk of depression (*Kang, Kim & Je, 2018*). One of the causes of depression may be the damage of acetylcholine (*Shytle et al., 2002*). Long-term consumption of carbonated beverages can cause oxidative stress in Wistar rat brain tissue, decrease the expression of antioxidant markers, reduce the expression of acetylcholinesterase (AChE) in serum and brain tissue, increase mRNA expression of dopamine D2 receptor (DD2R), and decrease 5-HTT expression (*El-Terras et al., 2016*; *Cermak et al., 2020*). *Lin et al. (2020)* found that traditional Chinese herbal tea could enhance the stress resistance (oxidative stress and heat stress) of nematodes, and significantly reduce fat deposition and age pigment accumulation. In nematodes, the accumulation of oxidative damage can affect its neuromotor function, while the accumulation of ROS in intestinal tract can affect the feeding behavior of nematodes (*Zhao et al., 2014*). This physiological process may provide an explanation on the influence of beverages on movement and feeding behaviors in worms as seen in this study.

## CONCLUSIONS

In conclusion, the results obtained here showed that though there wasn't any acute lethal effect of the 18 kinds of beverages on nematodes, but they had variant effects on the motor behavior and feeding behavior of worms. Different effects on the behavior of nematodes may be related to their different components or properties. It is worth noting that children are one of the main consumers of beverages, and their nervous system development is immature. The influence of long-term large amount of beverage intake on sports ability and nervous system is worth paying attention to. In the follow-up, it is necessary to increase indicators appropriately and possible molecular biomarker to further explore the mechanism of its potential health hazards.

## ACKNOWLEDGEMENTS

The authors would like to thank CGC for *E. coli* OP50 and *C. elegans* strains.

### Funding

This work was financially supported by the National Key Research and Development Program of China (No. 2019YFC1604901). The funders had no role in study design, data collection and analysis, decision to publish, or preparation of the manuscript.

### Grant Disclosures

The following grant information was disclosed by the authors:
The National Key Research and Development Program of China: No. 2019YFC1604901.

## Competing Interests

The authors declare there are no competing interests.

## Author Contributions

- Wenjing Zhang conceived and designed the experiments, performed the experiments, analyzed the data, prepared figures and/or tables, authored or reviewed drafts of the article, and approved the final draft.
- Nan Zhang performed the experiments, prepared figures and/or tables, and approved the final draft.
- Shan Zheng performed the experiments, prepared figures and/or tables, and approved the final draft.
- Wei Zhang performed the experiments, prepared figures and/or tables, and approved the final draft.
- Jingjing Liu performed the experiments, prepared figures and/or tables, and approved the final draft.
- Liwei He performed the experiments, prepared figures and/or tables, and approved the final draft.
- Anastasia Ngozi Ezemaduka analyzed the data, authored or reviewed drafts of the article, and approved the final draft.
- Guojun Li performed the experiments, authored or reviewed drafts of the article, and approved the final draft.
- Junyu Ning performed the experiments, authored or reviewed drafts of the article, and approved the final draft.
- Bo Xian analyzed the data, authored or reviewed drafts of the article, and approved the final draft.
- Shan Gao conceived and designed the experiments, authored or reviewed drafts of the article, and approved the final draft.

## Data Availability

The raw measurements are available in the Supplementary Files.

## Supplemental Information

Supplemental information for this article can be found online at http://dx.doi.org/10.7717/peerj.13563#supplemental-information.

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
