# Peer review of "Effects of commercial beverages on the neurobehavioral motility of Caenorhabditis elegans"

_PeerJ, doi:10.7717/peerj.13563_

## Round 0.1 · original submission · Major Revisions

Please address the concerns of all reviewers and amend your manuscript accordingly.

·

Basic reporting

• The manuscript was written in clear professional English language. The authors should go through the spell checks and errors. eg Line 48: appropriately ; line 276: compared

• The authors have given relevant literature and background information. Authors should cite the work by Cermk et al. Whole-organism behavioral profiling reveals a role for dopamine in state-dependent motor program coupling in C. elegans. eLife. 2020; 9: e57093.´which is a very relevant literature to the described work.

• Figures are relevant, well labelled and described. Yet the authors could give some vedio evidence or recording of the nematode behavior. Additionally, a schemata of the described worm behavior could greatly enhance clarity for the readers.

Experimental design

The work is interesting given the documented ill-effects of certain beverages to human population and is relevant to the scope of Peer J. Research questions are well defined as the authors want to evaluate the effects of 18 different beverages on behavioral modulation of nematodes and extrapolate on its toxicity. The experiments are performed well and methods described sufficiently.

Validity of the findings

Although the findings are interesting and useful, there are certain concerns that the authors need to address for publication in Peer J.

1. Figure 1 has bar charts with significance, please provide error bars in the graphs
2. In many places in the manuscript, (Line 213, Line 220) data is expressed as 12%~49% and 36%~43% respectively (P<0.05). Please express it as mean ± sem.
3. In the introduction, typically lines 66-71 comes at the end of introduction. The authors should shift this part to the end before starting the methods section. Instead authors could put a simple line here such as ´we proposed to study the effect of beverages on behavior of C.elegans´ and continue with the next paragraph.
4. The authors focus on the positive effects. They should also clearly conclusively state the negative effects and describe/discuss the reasons. For example: in this line 251: while fruit juice, carbonated drinks, relieve fatigue drink, functional beverages, tea and coffee all showed the characteristics of weakened feeding and mobility, which may be related to the food search process of nematodes. The authors should discuss the possible reasons for this result.
5. In line 233: they state that feeding behaviour are used to evaluate neurotoxicity. Could they discuss or evaluate the feeding behavior they recorded with neurotoxicity?
6. Lines 260, 266: It should be mentioned as 4a, 4b, 4c rather than 4abcd
7. Some page numbers in references are not proper. Example: line 414: 161-169 rather than 161-9. Same is true in line 423, 434. Please carefully check all references.

Reviewer 2 ·

Basic reporting

the aim of this study looks like to screen certain bioactivity of several commercial beverages using C. Elegans as the animal model, from this aspect, the novelty of the present study is not clear. The chemical composition of the samples should be provided, and further molecular biological studies need to be conducted to support behavioral observation results.

Experimental design

as the authors suggested, other endpoints such as reproduction, lifespan, molecular stress response, etc, should be included in the experimental design. And in my opinion, there was no logical correlation to compare the 18 beverages as their chemical components are totally different. this experimental design is more like an observational study, or a bioactivity screening.

Validity of the findings

no comment

·

Basic reporting

Manuscript is well written and sufficient context is provided.

Is there a reason why the bar charts in Figure 1 lack error bars?

In Table 1 dairy beverage section, please comment on the difference between prepared milk beverages A, B, C and D. Since their characteristics are very similar, what is the rationale for having so many milk beverage?

What does the heading ´proportion´ mean in Table 1? Please change to a meaningful heading or what information is the proportion values provide here?

Give the units along with the heading in Table 1. Eg: pascal for osmotic pressure

Experimental design

Research question is relevant and meaningful.

The authors must provide a detailed methodology of how they measured the beverage characteristics. A method section detailing how they measured osmotic pressure, specific gravity and the instruments used. This is critical for reproducibility.

Validity of the findings

Why are the LC50 values not provided ?

Line 261-262: based on figure 4 ABCD it has been mentioned that head thrash frequency is reduced 23%~30% and similar percentages are given based on Figure 4. It is not clear how this has been calculated as the figure does not seem to reflect these values.

---

## Round 0.2 · accepted · Accept

All issues pointed out by the reviewers were adequately addressed during revision and the amended manuscript is acceptable now.

·

Basic reporting

THE AUTHORS HAVE ADDRESSED ALL CONCERNS SUFFICIENTLY AND I RECCOMMEND PUBLICATION IN PeerJ

Experimental design

-

Validity of the findings

-

Additional comments

-

·

Basic reporting

no comment

Experimental design

no comment

Validity of the findings

no comment

Additional comments

The authors satisfactorily addressed all my comments.